# Effect of GO Additive in ZnO/rGO Nanocomposites with Enhanced Photosensitivity and Photocatalytic Activity

**DOI:** 10.3390/nano9101441

**Published:** 2019-10-11

**Authors:** Chatchai Rodwihok, Duangmanee Wongratanaphisan, Yen Linh Thi Ngo, Mahima Khandelwal, Seung Hyun Hur, Jin Suk Chung

**Affiliations:** 1School of Chemical Engineering, University of Ulsan, Daehak-ro 93, Nam-gu, Ulsan 680-749, Korea; c.rodwihok@hotmail.com (C.R.); ngoyenlinh0912@gmail.com (Y.L.T.N.); mahimaiitr@gmail.com (M.K.); shhur@ulsan.ac.kr (S.H.H.); 2Department of Physics and Materials Science, Faculty of Science, Chiang Mai University, Chiang Mai 50200, Thailand; duangmanee.wong@cmu.ac.th

**Keywords:** ZnO/rGO nanocomposites, UV detection, photocatalytics, morphological tunability, GO additive

## Abstract

Zinc oxide/reduced graphene oxide nanocomposites (ZnO/rGO) are synthesized via a simple one-pot solvothermal technique. The nanoparticle–nanorod turnability was achieved with the increase in GO additive, which was necessary to control the defect formation. The optimal defect in ZnO/rGO not only increased ZnO/rGO surface and carrier concentration, but also provided the alternative carrier pathway assisted with rGO sheet for electron–hole separation and prolonging carrier recombination. These properties are ideal for photodetection and photocatalytic applications. For photosensing properties, ZnO/rGO shows the improvement of photosensitivity compared with pristine ZnO from 1.51 (ZnO) to 3.94 (ZnO/rGO (20%)). Additionally, applying bending strain on ZnO/rGO enhances its photosensitivity even further, as high as 124% at r = 12.5 mm, due to improved surface area and induced negative piezoelectric charge from piezoelectric effect. Moreover, the photocatalytic activity with methylene blue (MB) was studied. It was observed that the rate of MB degradation was higher in presence of ZnO/rGO than pristine ZnO. Therefore, ZnO/rGO became a promising materials for different applications.

## 1. Introduction

Zinc oxide (ZnO) is a promising n-type semiconductor material that is used in a wide range of applications, such as gas detection [1,2,3,4], dye-sensitized solar cells [5,6,7], antibacterial surface coatings [8], light-emitting diodes (LEDs) [9,10], nanopower generators [11], ultraviolet (UV) detection [12,13,14,15], and photocatalytic applications [16,17,18]. ZnO nanostructures have a direct wide band gap (3.37 eV) [19], chemical stability [20], optical [21], piezoelectric [22,23,24], and electrical [25] properties. Additionally, ZnO possesses piezoelectric properties and self-carrier generation when tensile strain force is applied or substrates are bent [26]. ZnO based on UV detection is governed by photogenerated electrons that trap oxygen molecules at the surface of ZnO and change sensor resistance. In other way, ZnO as a photocatalyst, the photogenerated electron–hole pairs upon excitation may react with oxygen and water molecules resulting in free radicals that can degrade organic and inorganic compounds in aqueous medium. However, the performance of ZnO-based UV detection and photocatalytic activities faces some drawbacks in surface morphology, charge transportation, and recombination. Beside the above-mentioned drawbacks, when used as a photocatalyst, ZnO has very limited capability for visible light adsorption and can generate electron–hole pairs only in UV light, which contributes only 2–3% to total sunlight. Pati et al. found that the point defect concentration and optical/gas sensitivity of ZnO is due to the vacant lattices acting as active sites for oxygen trapping [27]. Zhang et al. reported that the existence of disorder and surface defects in ZnO crystals could improve the separation of photogenerated electron–hole pairs, preventing recombination and enhancing photocatalytic activity [28]. With the increase in lattice defects, the trapping of oxygen on the surface will increase, resulting in an improvement in UV sensing properties and prolonging the photocatalytic activity. In the case of ZnO nanoparticles, the potential barrier at the interface between the ZnO–ZnO grain boundaries results in slow charge transfer across grain boundaries and fast recombination. Some researchers reported that enhanced charge separation can provide a progression in interfacial charge transfer for dyes adsorption and photocatalytic degradation improvement [29,30]. To improve UV photosensitivity and photocatalytic activity, this carrier concentration, carrier pathway, and charge recombination should be modified [31].

In recent years, several efforts have been made to improve carrier concentration, to provide alternative carrier pathway and reduce charge recombination by tuning ZnO morphology, surface defect improvement [31,32,33,34,35], and combining ZnO with other functional materials, such as carbon-based graphene oxide (GO) [13] and reduced graphene oxide (rGO) [36,37]. GO, rGO, and graphene consist of sp${}^{2}$ hybridized carbon atoms arranged in two-dimensional honeycomb lattices. For UV detector applications, the interaction of ZnO and rGO could enhance UV sensing properties, in terms of inhibiting the recombination of electron–hole pairs and an increase in the photocurrent [38]. In addition to a capacity for photogenerated electron transportation from band gap excitation of semiconductors upon UV illumination and flexible mechanical properties that would support a self-carrier generator when bending strain is applied [39], applying tensile strain to ZnO/rGO hybrid nanostructures may improve photosensitivity in terms of increasing carrier density and carrier transfer. With regard to metal oxide based on photocatalytic activities, Zhang et. al. reported that graphene additive can improve photocatalytic activity by prolonging photogenerated electrons-hole pairs in TiO${}_{2}$, which efficiently suppresses the recombination of the photogenerated electron–hole pairs in TiO${}_{2}$ [40]. Pronay et al. reported that rGO laminated TiO${}_{2}$-B NW nanocomposites shows an excellent visible light dye degradation with superior degradation rate. This is due to defect creation and the narrowing of energy band gap, together with increasing adsorption by π-π interaction between GO and dye molecules [41].

Herein, we have demonstrated a simple one-pot solvothermal technique to prepare ZnO/rGO nanocomposites with varying the GO content. Specifically, the amount of GO content was varied to control the dimensionality of ZnO nanostructures and surface defects, which play important roles in photosensitivity and photocatalytic activity. The effects of tuned morphology on structural components, surface morphology, surface defects, and optical properties were also studied. As-synthesized ZnO/rGO was employed to evaluate UV photosensitivity assisted with applied bending strain and visible light photocatalytic activity of methylene blue (MB) degradation.

## 2. Experimental Details

### 2.1. Materials

Analytical-grade zinc acetate dihydrate (Zn(CH${}_{3}$COO)${}_{2}$·H${}_{2}$O, Samchun Pure Chemicals, Pyeongtaek, Korea), sodium hydroxide (NaOH, Samchun Pure Chemicals, Pyeongtaek, Korea), and commercial graphene oxide (GO, GO-V20-100, Standard Graphene, Ulsan, Korea) were procured and used without further purification. Deionized water (DI) was used to prepare solutions and wash all samples.

### 2.2. Preparation of ZnO/rGO Nanocomposites

ZnO/rGO was prepared by a solvothermal technique as shown in Figure 1a. First, 0.04 M of zinc acetate dihydrate was dissolved in 80 mL of DI under continuous stirring at room temperature for 30 min. to study the effects of GO additive, a precalculated amount of dispersed GO in DI (5 mg/mL) was prepared separately under ultrasonication, corresponding to 10, 20, and 30 wt% of GO. Then, the prepared GO solution was dropped into a zinc acetate dihydrate solution under continuous stirring for 30 min. After that, 1 g of NaOH in 50 mL of DI was added dropwise to the mixed solution until a pH of 10 was reached. Finally, the mixed solution was transferred to a Teflon-lined stainless steel autoclave under 150 ${}^{\circ }$C for 6 h. The product was collected by filtration and washed by DI several times and dried at 70 ${}^{\circ }$C for 6 h.

### 2.3. Characterization of ZnO/rGO Nanocomposite

Field-emission scanning electron microscopy (FE-SEM, JSM-600F, JEOL-JSM-7600F, Tokyo, Japan) and X-ray diffraction (XRD, Rigaku D/MAZX 2500V/PC model, Tokyo, Japan) were used to characterize the surface morphology and crystalline structure of ZnO/rGO. Fourier transform infrared spectroscopy (FT-IR, Thermo Electron Co., Waltham, MA, USA) and Raman spectroscopy (RS, Thermo Fisher Scientific, Waltham, MA, USA) were utilized at room temperature to investigate the functional groups and surface components of ZnO/rGO. Optical studies were considered the absorption spectra from 200 to 800 nm using an ultraviolet-visible spectrometer (UV-Vis, Spectramax Plus 384, Molecular Devices Co., San Jose, CA, USA). The surface properties and oxidation state of samples were investigated by X-ray photoelectron spectroscopy (XPS, K-alpha; Thermo Fisher Scientific Co., Waltham, MA, USA).

### 2.4. UV Sensing Measurement

To study UV sensing properties based on ZnO/rGO, an electrode was prepared on a transparent film (thickness: 100 μm). First, a transparent film substrate was cleaned in ethanol and DI and dried with a nitrogen gun. A UV sensor device based on ZnO/rGO was simply fabricated as a resistor in the electronic circuit. To make the sensing film, the as-synthesized ZnO/rGO powder was dissolved in DI (0.5 mg/mL) under ultrasonication. The mixed solution was then sprayed on the transparent film substrate and dried for 30 min at 60 ${}^{\circ }$C. The spray-coating step was repeated three times to ensure full coverage and film uniformity. The thickness of prepared film was shown in Appendix A. Finally, a silver paste was applied to both sides of the sensor device and kept in an oven for 6 h at 60 ${}^{\circ }$C. UV sensing properties based on ZnO/rGO were examined with a Blacklight blue lamp (EL20W BLB; 352 nm and 368 nm) with fixed intensity of 0.62 mW/cm${}^{2}$ at room temperature. As shown in Figure 1b, the as-prepared UV sensor sample was tested in UV testing box, with and without bending strain. To study bending strain effects, the prepared samples were placed on cylinders holders with radii of 12.5, 10, and 7.5 mm and tested in on/off UV illumination states. The UV sensing system was controlled by KickStart software (Tektronix Company, Beaverton, OR, USA) and Keithley 2400 Source Meter.

### 2.5. Evaluation of Photocatalytic Degradation

The photocatalytic activities of the samples were evaluated by degradation of MB as illustrated in Figure 1c. Prior to assessing the photocatalytic degradation activities, 10 mg of as-synthesized ZnO/rGO powder was added to an aqueous solution of MB (50 mL, 10 mg/L). The suspension was stirred for 30 min in the dark at 250 rpm to reach adsorption/desorption equilibrium. The prepared solutions were then illuminated by a 20W compact fluorescent lamp (EL20W; OSRAM dulux superstar). At specified time intervals, 2 mL of illuminated solutions was collected to record the absorption spectrum by UV–visible spectrometer. PL spectrum was performed to verify the recombination state of electron–hole pairs in samples (PL spectrometer with 473 nm diode laser; G9800A; Agilent Technologies, Santa Clara, CA, USA).

## 3. Results and Discussions

### 3.1. Structural Component, Surface Morphology, and Optical Properties

All structural components of the as-synthesized samples were confirmed by XRD as shown in Figure 2. The XRD patterns of GO and ZnO/rGO were recorded in the 2θ range of 5 to 80${}^{\circ }$. The XRD patterns of GO show peaks at 2θ values of 11.2${}^{\circ }$ corresponding to (002) [42]. The as-synthesized hybrid samples of ZnO, ZnO/rGO (10%), ZnO/rGO (20%), and ZnO/rGO (30%) exhibit prominent peaks at 2θ values of 31.8${}^{\circ }$, 34.4${}^{\circ }$, and 36.2${}^{\circ }$, corresponding to a ZnO wurtzite phase (JCPDS card of 89-13971) and indexed as (100), (002), and (101) reflection planes, respectively. From the XRD analysis, the lattice constants of ZnO with different GO contents can be calculated by a hexagonal space lattice equation:(1)$$\frac{1}{d^{2}}=\left(\frac{4}{3}\right)\left(\frac{h^{2}+hk+k^{2}}{a^{2}}\right)+\frac{l^{2}}{c^{2}}$$
where *d* is the interplanar distance and *h, k*, and *l* are miller indices. *a* and *c* refer to the lattice constant of the hexagonal structure. The size of a ZnO crystal can be calculated using Scherer’s equation [43]:(2)$$t=(\frac{K\lambda }{\beta \cos\theta })$$
where *t* is crystal size, *K* is a dimensionless shape factor, λ is the X-ray wavelength, β is the broadening at half of the maximum intensity, and θ is Bragg’s angle. Lattice constant and crystal size were calculated and summarized in Table 1. As GO content increased, the diffraction peak corresponded to the (002) reflection plan of ZnO intensified. This intense (002) peak plays a significant role in tunabilities of ZnO morphology in ZnO/rGO. However, the absence of a GO peak in ZnO/GO implied to a complete exfoliation and dispersion of GO into the ZnO [44,45].

The surface morphology of ZnO/rGO was observed by FE-SEM, as shown in Figure 3a–e. The figure demonstrates the morphology of GO and ZnO/rGO containing 0%, 10%, 20%, and 30% GO. The morphology of GO consisted of two-dimensional sheets with an average length of 89.3 μm. The average diameter of ZnO in ZnO/rGO containing the different amounts of GO is summarized in Table 1. Interestingly, the increase in GO additive did not change the diameter of ZnO, but tuned the morphology of ZnO in different as-synthesized ZnO/rGO. ZnO in the absence of GO took on a spherical shape with a diameter of 31.7 nm. However, ZnO in ZnO/rGO samples with 10 and 20% GO formed a rod shape along with spherical nanoparticles. At 30% GO, nanorods with a diameter of 24.6 nm and a length of approximately 106.8 nm could be seen. An increase in GO content is, therefore, a synergistic effect that controls dimensional ZnO formation through a tuning of the dimensional ZnO nanostructures from nanoparticles to nanorods, which corresponds to XRD results.

Figure 4a shows the FT-IR spectra of GO and as-synthesized samples. GO shows the presence of different peaks at 3450 cm${}^{-1}$, 1700 cm${}^{-1}$, 1620 cm${}^{-1}$, 1226 cm${}^{-1}$, and 1052 cm${}^{-1}$, corresponding to O–H stretching vibration of the surface-absorbed water molecules, carbonyl group (C=O), C=C aromatic configurable vibration, carboxyl C–O, and alkoxy C–O stretching, respectively [46]. The intensity of the peaks corresponding with oxygen functionalities in ZnO/rGO was reduced and some peaks (1226 cm${}^{-1}$ and 1052 cm${}^{-1}$) disappeared. This suggests that the majority of the oxygen functional groups in GO was reduced.

Raman spectra of GO and as-synthesized ZnO/rGO are shown in Figure 4b,c. The observed spectrum of GO exhibiting typical D and G bands is shown in Figure 4c. The D band at 1359 cm${}^{-1}$ is attributed a structural defect in the hexagonal graphitic layers, whereas the G band at 1597 cm${}^{-1}$ can be assigned to $sp^{2}carbon$-type structure [47]. As-synthesized ZnO/rGO exhibited an increased intensity ratio between the D and G peaks${}^{r^{**}}$ ($I_{D}/I_{G},$) compared with GO. In addition, the D and G band values shifted to lower frequencies as summarized in Table 2. Along with FT-IR observation, this suggests a reduction of oxygen functional groups in GO and restoration of new graphitic domains from the amorphous region of GO [48,49]. Moreover, the Raman spectrum of pure ZnO exhibited optical phonon vibrational peaks corresponded to E${}_{2}\left(high\right)$-E${}_{2}\left(low\right)$, A${}_{1}\left(TO\right)$, E${}_{2}\left(high\right)$, and E${}_{1}\left(LO\right)$ vibrational modes (Figure 4b). The most intense peak, E${}_{2}\left(high\right)$, is associated with intrinsic stress of ZnO wurtzite at 435.84 cm${}^{-1}$. The E${}_{1}\left(LO\right)$ peak at 566.79 cm${}^{-1}$ is attributed to surface defect formation such as dislocation and oxygen-zinc vacancy states [19]. As shown in Table 2, the intensity ratio of the E${}_{1}\left(LO\right)$ peak to the E${}_{2}\left(high\right)$ peak${}^{r^{*}}$ increased in the as-synthesized ZnO/rGO at different GO additive and reached a maximum at 0.686 in ZnO/rGO (20%) sample. This indicates that the most defect/disorders occurs in the ZnO/rGO (20%) sample, compared with the other as-synthesized ZnO/rGO samples and GO compared with other as-synthesized ZnO/rGO with different amounts of GO content. The optimization of adequate amounts of GO is, therefore, necessary to control defect formation in ZnO/rGO.

The optical properties of ZnO/rGO were studied by UV–visible spectroscopy. Figure 5a shows the absorption spectra of ZnO/rGO synthesized with different amounts of GO. The optical absorption coefficient (α) can be calculated using Equation (Equation 3) [50]:(3)$${\left(\alpha h\nu \right)}^{2}=D\left(h\nu -E_{g}\right)$$
where *D* is constant, hν is the incident photon energy and $E_{g}$ is the optical band gap. According to Equation (Equation 3), the band gap of materials has been calculated by conventional extrapolation of the plot of ${\left(\alpha h\nu \right)}^{2}$ and hν at α = 0, as shown in Figure 5b. The optical band gap value of as-synthesized ZnO/rGO is summarized in Table 2. The optical band gap decreased with an increase in the amount of GO content. For comparison purposes, the band gap energy of pristine ZnO shows higher value (3.143) compared with as-synthesized ZnO/rGO samples (2.897–2.976). The decrease in the band gap of ZnO/rGO can be explained by the increase in the surface charge between ZnO and rGO, which results in the optical band gap shifting to a higher wavelength. This is similar to a result obtained by R. Paul et al. [13]. In addition, the presence of defects in the as-synthesized ZnO/rGO samples can be determined using Urbach absorption tail energy [51,52] by
(4)$${\left(E_{u}\right)}^{-1}=\frac{\ln\alpha }{h\nu }$$
where $E_{u}$ is the Urbach absorption tail energy, which is assumed to be the width of the exponential edge. The Urbach absorption tail energy shows a smearing of absorption edge, supplying information on the presence of impurities or defects in the as-synthesized ZnO/rGO. A close examination of the results reveals that an increase in GO additive increases the value of $E_{u}$ to certain GO content, after which the $E_{u}$ value decreases. The maximum value of $E_{u}$ was obtained at a GO content of 20%, indicating to the highest presence of defects/disorder in ZnO/rGO synthesized at 20% GO. These results confirm the existence of defect in ZnO/rGO assisted with Raman data, which also showed the highest values of the E${}_{1}\left(LO\right)$ peak to the E${}_{2}\left(high\right)$ peak ratio at ZnO/rGO synthesized at 20% GO.

ZnO/rGO samples were subjected to a X-ray photoelectron spectrometer to quantify their oxidation state, oxygen vacancy, and surface chemical composition, as shown in Figure 6a–c. Figure 6a shows the XPS survey spectra of all-synthesized ZnO/rGO, pristine ZnO, and GO with the presence of C, O, and Zn elements. The deconvoluted C1s peaks were fitted to three Gaussian peaks, consisting of $sp^{2}$/$sp^{3}$ carbon (C-C/C=C, 284.35–284.95 eV), C–OH/C–O/C–Zn (286.98–286.42 eV), and C=O/COOH (287.5–289) [46,47,53], as shown in Figure 6b. It can be clearly seen that the peak intensity of oxygen bonded carbon was decreased after the hydrothermal process. This suggests that the reduction of oxygen functional groups in GO. Figure 6c shows the deconvoluted O1s of ZnO/rGO compared with pristine ZnO. The O1s peak was deconvoluted into three Gaussian peaks, corresponding to O${}_{L}$, O${}_{V}$, and O${}_{C}$ [54,55,56]. The O${}_{L}$ peak at 530.80–531.00 eV is been attributed to lattice oxygen ion (O${}_{2}^{-}$) in the ZnO wurtzite structure. The O${}_{V}$ peak at 531.53–531.95 eV has been assigned to O${}_{X}^{-}$ ion oxygen deficient region (oxygen vacancy) and the O${}_{C}$ peak at 532.47–533.20 eV corresponds to the chemisorbed oxygen on the surface. The % area and peak position of O 1s peak was summarized in Table 3. As mentioned above, the O${}_{V}$ peak is related to the oxygen vacancy which the intensity of O${}_{V}$ peak can be connected to the concentration of the oxygen vacancy in ZnO/rGO. In Figure 6c, the intensity of O${}_{V}$ peak increased with an increase in GO additive. The ratio of O${}_{V}$/O${}_{L}$ was used to compare the presence of oxygen vacancy and oxygen lattice in materials [57]. As seen in Table 3, the O${}_{V}$/O${}_{L}$ ratio shows the highest value in the ZnO/rGO (20%) sample, suggesting that oxygen vacancy is intensively constructed in ZnO/rGO (20%) more than other conditions and pristine ZnO. Along with Raman and UV–Vis results; this is in good agreement to confirm that the GO additive has an effect on oxygen vacancy formation in ZnO/rGO.

### 3.2. UV Sensing Measurement

UV sensor devices based on as-synthesized ZnO/rGO at different amounts of GO content were fabricated and tested at room temperature. The UV sensing measurement was investigated by using a UV illumination source with a wavelength of 365 nm and 2 V of bias voltage. The current characteristic was measured in “on” and “off” states for the photocurrent and dark current, respectively, over several cycles. To minimize the effect of natural light, all measurements were conducted in a dark box and the illumination time was fixed at 30 s. Photosensitivity was defined by the ratio of photocurrent and dark current (S = $I_{UV}/I_{D}$) as a function of time. Photosensitivity was calculated and compared for each condition, as summarized in Table 4. The time interval which the current increased to 90% of the maximum value (response time) was also determined. The UV sensor device performance was studied with and without applying bending strain.

Current–time (I–t) and current–voltage (I–V) characteristics based on ZnO/rGO without a bending effect are shown in Figure 7a,b. As seen in Figure 7b, the I–V curve shows the characteristics of ohmic contact between ZnO/rGO and a silver electrode. Pristine ZnO shows photosensitivity lower than as-synthesized ZnO/rGO as exhibited in Table 4. This indicates that GO additive can improve photosensitivity. Figure 7a shows how the photocurrent increased with an increase in GO content to 20% and then decreased as GO content rose to 30%. This is due to the declined ZnO and cut-off light of the active area of ZnO, leading to a decrease in photocurrent [58]. An increase in GO additive is therefore a synergistic parameter to enhance the photosensitivity, in respect of increasing carrier concentration from surface disorder and electron adsorption on the surface.

The UV sensing characteristics were demonstrated by changing several bending radii, as shown in Appendix A. For the UV sensing device based on pristine ZnO, photosensitivity increased when applying a 12.5 mm bending radius from 1.51 to 1.82. A decrease in bending radius resulted in a decrease photosensitivity. Similar to the as-synthesized ZnO/rGO, photosensitivity showed the same trend, which was optimized with an applied bending radius of 12.5 mm (Table 4). The comparison of photosensitivity of as-synthesized ZnO/rGO at a different applied bending radius is depicted in Appendix A. As-synthesized ZnO/rGO (20%) showed the maximum photosensitivity (8.81), which was improved by 123.59% compared with free-applied bending (Figure 7c). The as-synthesized ZnO/rGO (30%) sample showed an improvement in photosensitivity by 244.57%, with an applied 12.5 mm bending radius. However, decreasing bending radius below 10 mm led to photosensitivity reduction.

The improvement in the photosensitivity of ZnO/rGO with and without bending strain can be explained in a schematic model, as shown in Figure 8. Typically, the photosensitivity of a UV detector mechanism based on n-type ZnO nanostructures can be described by the chemisorption of oxygen. Without UV illumination, oxygen molecules adsorb to the surface, resulting in a decrease in surface conductivity, which can be expressed by O${}_{2}\left(g\right)+e^{-}\to O_{2}^{-}\left(ads\right)$, as shown in Figure 8a left side. Upon UV illumination, electron–hole pairs are generated by $h\nu \to e^{-}+h^{+}$. Adsorbed oxygen ions will be discharged by holes to produce oxygen molecules and release trapped electrons to the conduction channel, resulting in an increase of surface conductivity, which can be expressed as O${}_{2}^{-}\left(ads\right)+h^{+}\to O_{2}\left(g\right)$, as shown in the right side of Figure 8a. The photosensitivity performance depends on the surface-to-volume ratio and surface defects (oxygen vacancies and/or zinc interstitials), which are regularly responsible for carrier concentration. The more defects exist in the sample, the more carriers will increase.

For the bending effect, as shown in Figure 8b, the possibility of improved photosensitivity can be described in two aspects. First, with incremental surface defects [59,60], the applied bending on ZnO/rGO will expand the surface area and increase the surface defect between ZnO/rGO-ZnO/rGO interfaces. More oxygen could therefore travel to rGO sheet and increase oxygen adsorption, leading to improvement in photosensitivity under UV illumination, as shown in Figure 8b. A decrease in bending radius did not only increase oxygen adsorption on the sensing surface, but also resulted in collateral destruction (cracking) of the structural sensing surface and contact between the sensing surface and the silver electrode (in the case of application of a bending radius over the elastic limit of materials). This would result in carrier pathway deformation, carrier mobility, and photosensitivity reduction. However, an appropriate bending radius could increase oxygen adsorption and not result in cracking on the sensing surface; this would improve photosensitivity. The second aspect is the piezoelectric effect [14,26]; this is due to the growth of ZnO, which adhered to the rGO surface, assuming that rGO can transfer all of its bending force to ZnO nanostructures. When bending strain was applied to the substrate, a piezoelectric charge was induced on the ZnO surface, leading to an increase in adsorbed oxygen on the surface and producing a larger depletion region, as shown in Figure 8b, left side. Under UV light, oxygen desorption released electrons back to the conduction channel, leading to the observed increased photosensitivity as shown in Figure 8b right side. The present research found that an increase in the GO additive can improve the photosensitivity of a UV sensing device based on ZnO/rGO, in terms of induced intrinsic disorder/oxygen vacancy formation in ZnO. Additionally, a UV sensing device with applied bending exhibits superior photosensitivity in respect of more surface area and surface defect creation, providing more absorbed oxygen and more induced negative piezoelectric charge from piezoelectric effect. We also compared the photosensitivity in our work with previously reported research in Appendix A. This can be clearly seen that as-synthesized ZnO/rGO has good photosensitivity with faster response time compared with other reports.

### 3.3. Photocatalytic Activity Studies

The photocatalytic activity of ZnO/rGO was measured in terms of the degradation of MB under visible light. The MB absorption band was observed at 664 nm. The change in the absorption peak of MB was used to measure variation in MB concentration with a visible light irradiation time of 0, 5, 10, 20, 40, and 60 min as shown in Appendix A. Figure 9a shows the variation in the absorption band of the MB solution with the presence of ZnO/rGO sample as a function of visible light irradiation time. The photodegradation kinetic model can be explained by a generic modal equation as a logarithm of time and MB concentration, as given by
(5)$$\ln\frac{C_{t}}{C_{0}}=-k_{c}t$$
where $C_{0}$ and $C_{t}$ are the absorption of MB solution before and after visible light irradiation, respectively; $k_{c}$ is photocatalytic reaction rate constant; and *t* is irradiation time. From Figure 9a, it can be seen that the concentration of MB has been reduced with increasing irradiation time, indicating the photodegradation of MB in the presence of ZnO/rGO. Additionally, the photodegradation of MB in ZnO/rGO was considerably greater than in pure ZnO. Appendix A shows that ZnO/rGO nanocomposite with 20% GO exhibited the highest MB photodegradation activity (93.78%) within 60 min under visible light compared with 30% GO (86.43%), 10% GO (56.62%), and without GO content (4.11%). A comparison of the as-calculated degradation rate constants is shown in Table 4. This result reveals that ZnO/rGO with 20% GO had the highest photocatalytic activity, with a rate constant of 0.0482 min${}^{-1}$, which is almost 18 times greater than pristine ZnO (0.0027 min${}^{-1})$ and other as-synthesized ZnO/rGO samples. As a result, we can draw a conclusion that GO additive plays an important role in controlling photocatalytic activities by induced-defect/oxygen vacancy formation in ZnO. Furthermore, it provides an alternative carrier pathway and separates photogenerated electron–hole pairs preventing charge carrier recombination. Photodegradation of MB in this work is compared with previously reported literatures in Appendix A. It can be seen that as-synthesized ZnO/rGO shows better photodegradation rate (%) of MB with shorter UV illumination time.

The photocatalytic activity mechanism by ZnO/rGO is illustrated in Figure 10. ZnO acted as an electron–hole pair generator, and rGO acted as the carrier pathway. The as-synthesized ZnO/rGO was excited to generate electron–hole pairs of ZnO by visible light illumination. Then, the photogenerated electrons move up freely to the conduction band of ZnO and transfer to the nearest rGO surface. Additionally, in the ZnO/rGO heterojunction, the photogenerated electron in ZnO can freely move to rGO, as the work function of rGO is lower than the conduction band level of ZnO. This will provide alternative carrier pathway and enhance charge separation which have a beneficial effect for photocatalytic [61]. The photogenerated electrons and holes reacted with oxygen and hydroxyl groups to form superoxide radicals and hydroxyl radical ions. The generated radicals then degrade MB and produce oxidized organic products. The observed enhanced photocatalytic activity can be attributed to the surface properties of nanocomposites that provide a high surface area, enhancing MB absorption [62]. Photodegradation abilities generally depend on photogenerated electron–hole pairs. In the case of pristine ZnO with few disorder and surface defects, the photogenerated electrons and holes undergo quick recombination, leading to a lower concentration of photogenerated electron–hole pairs compared with ZnO/rGO with high disorder and surface defects (as evaluated by Raman spectra, Urbach energy, and XPS).

To additionally support the charge recombination information, a room-temperature PL spectrometer with excitation wavelength at 325 nm was performed to show up the charge recombination of ZnO and as-synthesized ZnO/rGO. As shown in Appendix A, the PL spectra of all the as-prepared samples show a sharp emission peak in the UV region of 380 nm, which is related to the band-to-band radiative recombination of the photogenerated electron–hole. Additionally, three minor, broad emission peaks in the visible region of 480 to 625 nm were also found, corresponding to defects of ZnO, such as oxygen vacancy and zinc vacancy. No detectable emission peak shift was found in any sample. However, luminescence quenching of as-synthesized ZnO/rGO was observed, in the order of ZnO/rGO (20%) > (30%) > (10%) > pristine ZnO. This suggests to ZnO/rGO (20%) has lower charge carrier recombination compared with other ZnO/rGO samples and pristine ZnO. Regarding the Raman, UV visible, and XPS analysis, the lower charge carrier recombination may be due to the existence of more defect/oxygen vacancy. Additionally, the GO additive provided the additional carrier pathway, which can increase charge separation efficiency and prolong photocatalytic reaction lifetime [28,63,64].

## 4. Conclusions

A simple one-pot solvothermal technique to prepare ZnO/rGO nanocomposites with high photosensitivity and high photodegradation of MB has been demonstrated. This study indicates that using the optimal amount of GO additive plays a significant role in not only nanoparticle–nanorod tunability, but also in inducing the intrinsic defect formation of ZnO. The ZnO/rGO based on UV detection and photocatalyst outperformed pristine ZnO, as the GO additive generates more defects and disorder, leading to increase absorbed oxygen for carrier concentration increment. Additionally, it provides the alternative charge carrier pathway to prevent charge carrier recombination and prolong photocatalytic reaction lifetime. ZnO/rGO (20%) shows the highest photosensitivity of 3.94 with 18.16 s of response time and degrades 93.78% of MB in 1 h. This makes it a promising candidate on double application of environmental monitoring and pollutant treatment.

## Figures and Tables

**Figure 1 nanomaterials-09-01441-f001:**
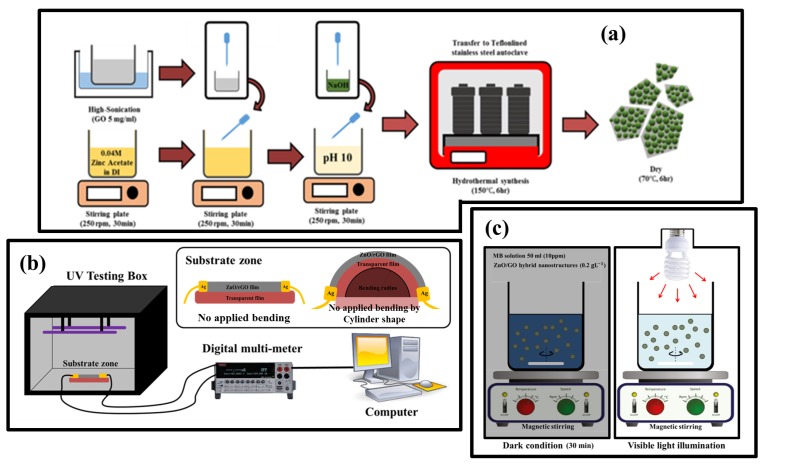
(**a**) The preparation process of ZnO/rGO nanocomposites, (**b**) UV sensing measurement, and (**c**) photocatalytic experiment.

**Figure 2 nanomaterials-09-01441-f002:**
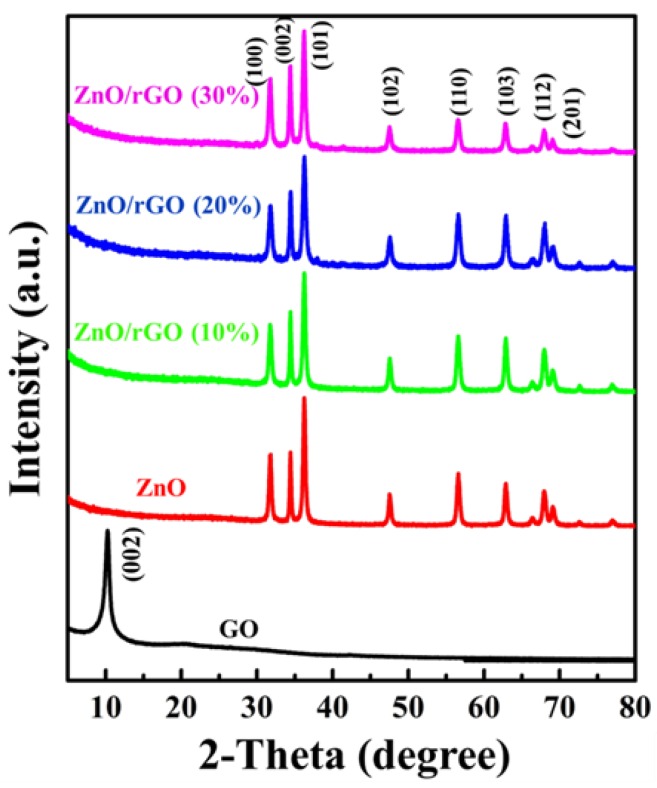
Pattern of GO and as-synthesized ZnO/rGO.

**Figure 3 nanomaterials-09-01441-f003:**
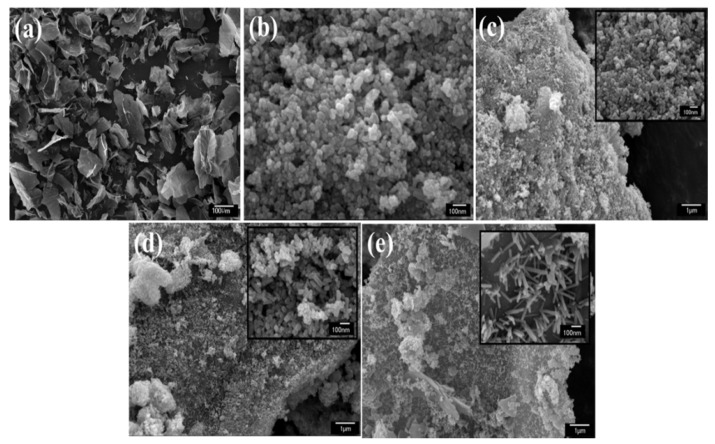
Field-emission scanning electron microscopy (FE-SEM) images of the (**a**) GO sheet, (**b**) ZnO, (**c**) ZnO/rGO (10%), (**d**) ZnO/rGO 20%, and (**e**) ZnO/rGO ZnO/rGO (30%). The inset shows the high magnification of FE-SEM image for each condition.

**Figure 4 nanomaterials-09-01441-f004:**
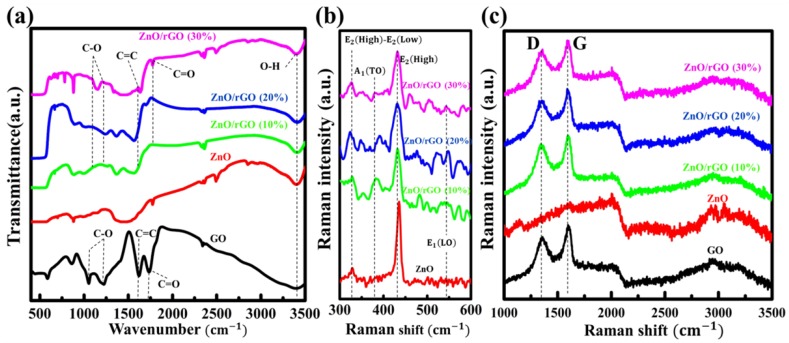
Surface characterization of ZnO/rGO. (**a**) FT-IR spectra and Raman spectra, (**b**) ZnO component, and (**c**) GO component.

**Figure 5 nanomaterials-09-01441-f005:**
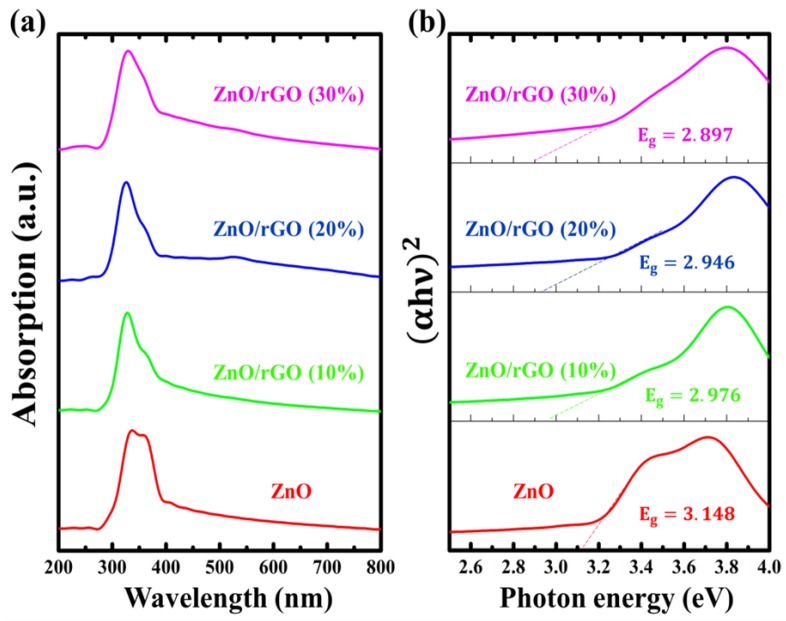
UV–visible spectroscopy of as-synthesized ZnO/rGO: (**a**) absorption spectra and (**b**) energy band gap.

**Figure 6 nanomaterials-09-01441-f006:**
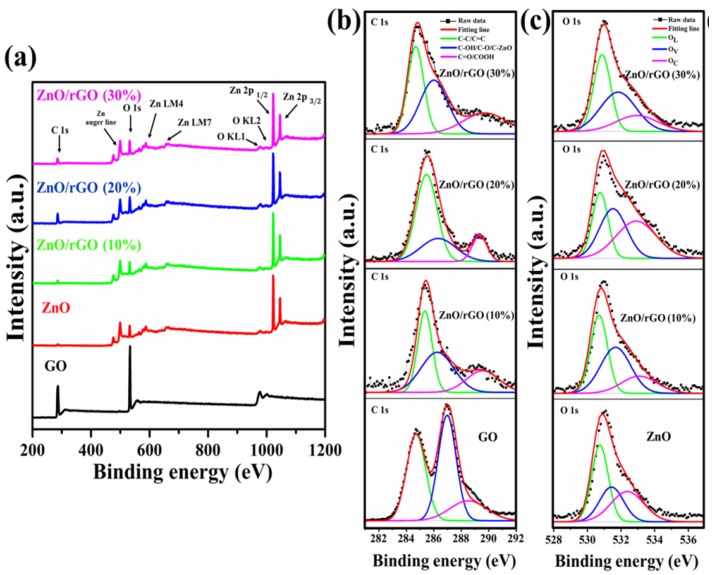
(**a**) The X-ray photoelectron spectroscopy (XPS) survey spectra obtained for GO. The deconvoluted (**b**) C 1s and (**c**) O 1s of GO, pristine ZnO, and as-synthesized ZnO/rGO.

**Figure 7 nanomaterials-09-01441-f007:**
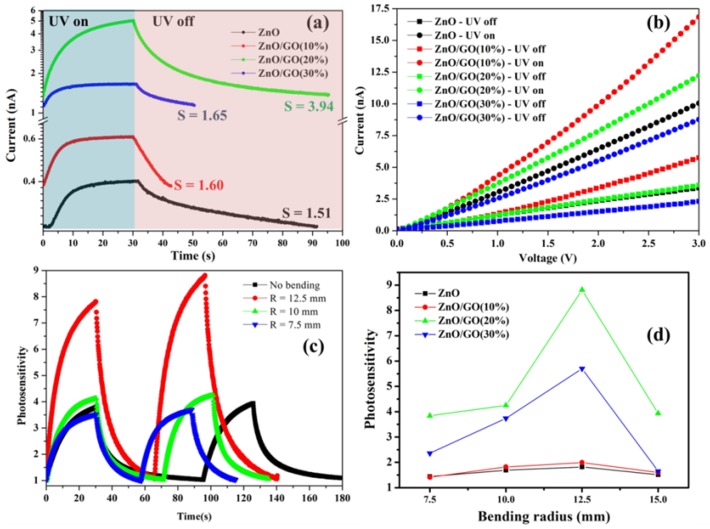
(**a**) Current–time (I–t) of ZnO/rGO, (**b**) I–V characteristic of as-synthesized ZnO/rGO without bending strain, and (**c**) current–time (I–t) of ZnO/rGO (20%) with bending radius and (**d**) comparison of photosensitivity.

**Figure 8 nanomaterials-09-01441-f008:**
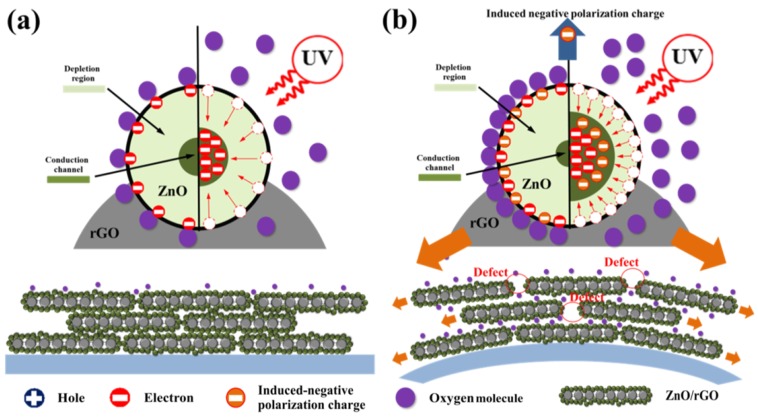
UV sensing mechanism of as-synthesized ZnO/rGO (**a**) without and (**b**) with bending strain.

**Figure 9 nanomaterials-09-01441-f009:**
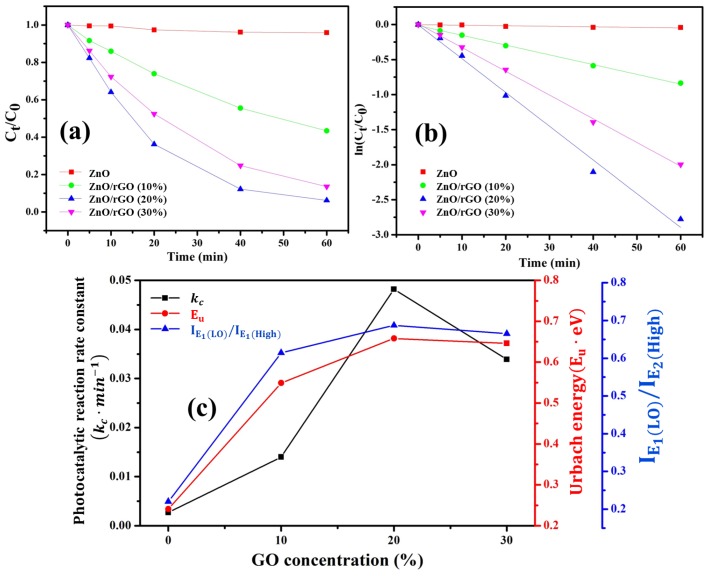
(**a**) Photodegradation of methylene blue (MB) in the presence of various photocatalysts under visible light illustration, (**b**) apparent first-order kinetics of photocatalytic degradation of MB, and (**c**) the effect of GO concentration on photocatalytic reaction rate constant and defect evaluation values.

**Figure 10 nanomaterials-09-01441-f010:**
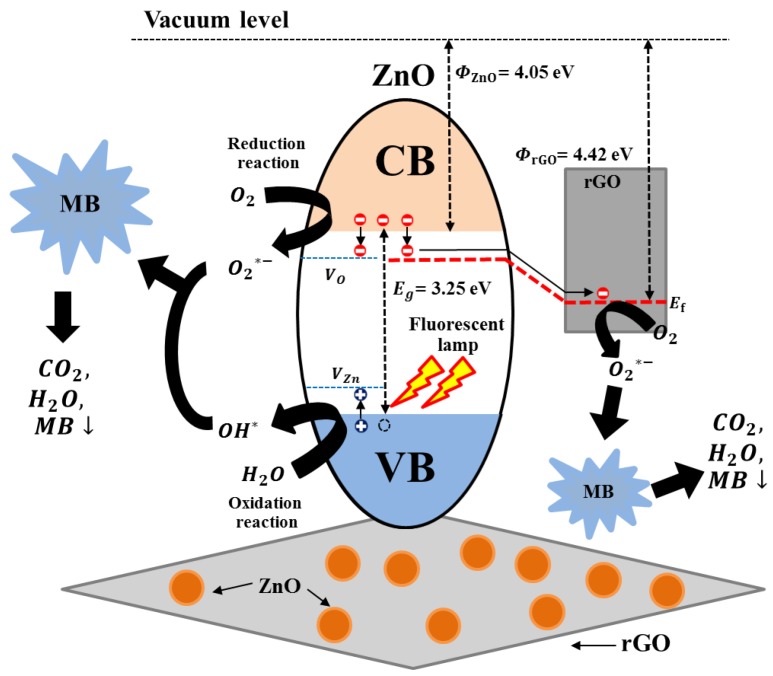
Photo-charge generation and charge transfer for MB degradation using as-synthesized ZnO/rGO reproduced with permission from [61], Copyright RSC Adv., 2014.

**Table 1 nanomaterials-09-01441-t001:** Average diameter, length, crystal size, and lattice parameter of the GO and as-synthesized samples.

Samples	SEM			XRD		
	Diameter (nm)	Length (nm)	Shape	Crystal Size (nm)	Lattice Parameter (Å)	
					a	c
ZnO	31.7	-	Nanoparticles	33.77	3.299	5.209
ZnO/rGO (10%)	25.8	44.1	Nanoparticles/ Nanorods	30.66	3.299	5.207
ZnO/rGO (20%)	23.8	63.8	Nanoparticles/ Nanorods	24.92	3.299	5.203
ZnO/rGO (30%)	24.6	106.8	Nanorods	25.62	3.300	5.209

**Table 2 nanomaterials-09-01441-t002:** Various optical and defect parameters of GO and as-synthesized samples.

Samples	Raman Data						UV–Vis Data	
	ZnO Vibrational Mode (cm${}^{-1}$)			GO Vibrational Mode (cm${}^{-1}$)			E${}_{g}$ (eV)	E${}_{u}$ (eV)
	E${}_{2}\left(\mathrm{high}\right)$	E${}_{1}\left(\mathrm{LO}\right)$	$r^{*}$	D Peak	G Peak	$r^{**}$		
GO	-	-	-	1358.61	1596.77	0.815	-	-
ZnO	435.84	566.97	0.219	-	-	-	3.148	0.241
ZnO/rGO (10%)	433.97	570.83	0.613	1343.18	1594.70	0.870	2.976	0.549
ZnO/rGO (20%)	431.98	566.01	0.686	1351.86	1588.10	0.867	2.946	0.658
ZnO/rGO (30%)	434.87	578.54	0.664	1357.66	1588.10	0.862	2.897	0.646

$r^{*}$: The intensity ratio of the E${}_{1}\left(LO\right)$ peak to the E${}_{2}\left(high\right)$ peak. $r^{**}$: The intensity ratio between the D and G peaks.

**Table 3 nanomaterials-09-01441-t003:** Atomic percentage of fitted high-resolution spectra of the O 1s of ZnO/rGO nanocomposites and pristine ZnO.

Sample	XPS – O 1s						
	O${}_{L}$		O${}_{V}$		O${}_{C}$		O${}_{V}$/O${}_{L}$ Ratio
	Peak Position	% Area	Peak Position	% Area	Peak Position	% Area	
ZnO	530.83	41.30	531.53	26.67	532.47	32.03	0.646
ZnO/rGO (10%)	530.82	40.08	531.80	41.50	533.20	18.42	1.035
ZnO/rGO (20%)	530.89	26.79	531.64	34.78	533.02	38.43	1.298
ZnO/rGO (30%)	531.00	38.84	531.95	41.50	533.05	19.66	1.068

**Table 4 nanomaterials-09-01441-t004:** Photosensitivity (increase/decrease percentage) of UV sensing and photocatalytic properties based on ZnO/rGO nanocomposites.

Samples	Photosensitivity (↑↓%)				Photocatalytic Activity		
	No Bending	Bending Radius (R${}_{b}$)			MB Degradation	k${}_{c}$ (min${}^{-1}$)	R${}^{2}$
		12.5 (mm)	10 (mm)	7.5 (mm)			
ZnO	1.51	1.82 (↑21.47%)	1.69 (↑12.70%)	1.44 (↓3.77%)	4.11%	0.0027	0.8793
ZnO/rGO (10%)	1.60	1.99 (↑23.55%)	1.82 (↑13.20%)	1.41 (↓12.21%)	56.62%	0.0140	0.9982
ZnO/rGO (20%)	3.94	8.81 (↑123.59%)	4.25 (↑7.85%)	3.84 (↓2.45%)	93.78%	0.0482	0.9901
ZnO/rGO (30%)	1.65	5.70 (↑244.57%)	3.74 (↑126.22%)	2.36 (↑42.83%)	86.43%	0.0339	0.9985

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
