# Peer review of "Effect of GO Additive in ZnO/rGO Nanocomposites with Enhanced Photosensitivity and Photocatalytic Activity"

_nanomaterials, 2019, doi:10.3390/nano9101441_

Round 1

Reviewer 1 Report

In this manuscript, the authors present the synthesis and characterization of composite materials comprised of zinc oxide and reduced graphene oxide in photodetection and photocatalysis. Combining ZnO with materials to improve the carrier concentration and other properties is a popular topic among researchers, and while ZnO/rGO materials have been reported previously for this purpose, the work here introduced a somewhat different synthetic approach that may be simpler to implement that alternatives while also offering tunability of the rGO content. Generally, the conclusions are supported by the evidence provided and the paper is of sufficient interest to warrant publication. There are, however, a few issues that would need to be addressed prior to publication:

- The introduction would benefit from a brief description of the importance of interfacial properties of materials used in water treatment applications (J. Appl. Phys. 124 (2018) 030901).

- ZnO has a significant drawback in photocatalysis, namely, that is only sensitive to UV photons, which are generally more expensive to generate. The topic of visible-light sensitivity in photocatalysis is an important point that could use more discussion in the manuscript (see, for example, Adv. Sust. Sys. 1 (2017) 1600041).

- The timescale for the photosensitivity seems rather slow given the timescale of charge dynamics in the material. Can the authors provide some explanation for why the photocurrent requires 10s of seconds to saturate (and to decay) rather than being nearly instantaneous? Presumably, defects in the material are playing a significant role.

- Regarding the methodology for the catalysis experiment, the way the samples were collected may influence the conclusions. Specifically, repeatedly removing 5 mL aliquots from a 50 mL solution will significantly alter the optical density of the overall solution. It would be helpful if the authors could include this variation in their analysis of the photocatalytic degradation performance.

- Typo on page 1: “dye-sensitized solar cells” rather than “dye-synthesized solar cells”

Reviewer 2 Report

GENERAL COMMENTS

G.1 This is a systematic piece of experimental work. 

G.2 The novelty of the research presented is rather limited since ZnO is a much-studied semiconductor oxide and rGO has been used as an additive to many photocatalytic/electro-photocatalytic systems). Apparently, the only novelty of this work has been the study of the effect of rGO content on UV response and photocatalytic activity; this and any other original parts need to be stressed an put in perspective with other developments in the field.

G.3 A more detailed description of experimental conditions is needed.

G.4 Based on the above as well as on the detailed comments presented below, the paper can be re-considered after MAJOR REVISIONS.

Detailed Comments

D.1 Please provide experimental details on how was GO transformed to rGO (and to what extent).

D.2 What is the transparent substrate used?

D.3 How thick was the ZnO/rGO film prepared? (Surely not 100 m as stated in 2.4)

D.4 Please provide a schematic description of thephoto-electrochemical device used. Was the substrate between the two layers conducting?

D.5 Please provide details of the UV lamp used; please comment whether its emmission spectrum had a tail overlapping with the spectrum of MB absorption.

D.6 Figure 10. Please add the Fermi level of rGO. On many occasions the semiconductor/rGO heterojunction is reported to have a beneficial effect on the photoresponse.

D.7 Please check throughout for language mistakes/ inaccuracies: e.g. "recombination prolongation"(recombination delay?) ; line 18: "dye synthesized solar cells" should be dye-sensitized etc.

Round 2

Reviewer 1 Report

The authors have made a number of minor revisions to address the comments raised during the initial review.  This work is now suitable for publication.

Reviewer 2 Report

The revised version of the paper can now be accepted for publication.